# WHY DOES THE EFFECTIVE CONTEXT LENGTH OF LLMS FALL SHORT?

**Chenxin An**[1] *  **Jun Zhang**[2]  **Ming Zhong**[3]  **Lei Li**[1]  **Shansan Gong**[1]
**Yao Luo**[2]  **Jingjing Xu**[2]  **Lingpeng Kong**[1]
[1]The University of Hong Kong  [2]ByteDance Inc.  [3]University of Illinois Urbana-Champaign

## ABSTRACT

Advancements in distributed training and efficient attention mechanisms have significantly expanded the context window sizes of large language models (LLMs). However, recent work reveals that the effective context lengths of open-source LLMs often fall short, typically not exceeding half of their training lengths. In this work, we attribute this limitation to the left-skewed frequency distribution of relative positions formed in LLMs pretraining and post-training stages, which impedes their ability to effectively gather distant information. To address this challenge, we introduce Shif̲T̲ed R̲otray position embedd̲ING̲ (STRING). STRING shifts well-trained positions to overwrite the original ineffective positions during inference, enhancing performance within their existing training lengths. Experimental results show that without additional training, STRING dramatically improves the performance of the latest large-scale models, such as Llama3.1 70B and Qwen2 72B, by over 10 points on popular long-context benchmarks RULER and InfiniteBench, establishing new state-of-the-art results for open-source LLMs. Compared to commercial models, Llama 3.1 70B with STRING even achieves better performance than `GPT-4-128K` and clearly surpasses Claude 2 and Kimi-chat. All code and data used in this work are released at https://github.com/HKUNLP/STRING.

## 1 INTRODUCTION

The increase in context length for large language models (LLMs; OpenAI 2023; Anthropic 2023; Bai et al. 2023; Xiong et al. 2023; Llama Team 2024) has facilitated the development of a wide range of applications (Pang et al., 2022; Bairi et al., 2023), substantially expanding the capabilities of AI systems. Recent advancements in efficient training and attention calculation (Li et al., 2024a; Dao, 2023; Liu et al., 2023) have made it feasible to train LLMs with exceptionally long context windows. For instance, Llama3.1 (Llama Team, 2024) features a context length of 128K tokens, which is $64\times$ longer than that of its initial release (Touvron et al., 2023a).

This trend towards longer context lengths in LLMs promises enhanced capabilities. Previous work has primarily focused on extending the context length of LLMs, with significant efforts devoted to improving data engineering techniques (Fu et al., 2024b; Hu et al., 2024; Bai et al., 2024; Zhao et al., 2024). High-quality natural long-context data are scarce in real-world settings, limiting the availability of such data for training purposes. To address this challenge, recent methods aim to generate synthetic training data that better capture the nuances of naturally occurring long-context information, despite inherent challenges such as time consumption in continual training and potential biases (Zhao et al., 2024; An et al., 2024b; Lv et al., 2024). Researchers have also focused on addressing specific architectural limitations. Efforts have been made to correct the improper adjustment of the base frequency in Rotary Position Embedding (RoPE) (Su et al., 2022; Peng et al., 2023; Chen et al., 2023; Lin et al., 2024b; Chen et al., 2024).

However, recent studies (An et al., 2023; Zhang et al., 2024d; Li et al., 2024b; Wang et al., 2024a) reveal a notable discrepancy between these theoretical improvements and observed performance. In practice, the effective context utilization of these models often falls substantially below their claimed or training context lengths. For example, on the widely used RULER benchmark (Hsieh et al., 2024),

---

*Work done during internship at ByteDance Inc.

the effective context length of the latest Llama 3.1 70B model is only 64K, despite employing scaled RoPE base frequency (Peng et al., 2023) and having sufficient training data (Llama Team, 2024). In fact, most open-source models demonstrate an effective context length less than 50% of their training length (Hsieh et al., 2024). A key research question emerges from these observations: *Why does the effective context length of LLMs fall short of their training context lengths?*

In this study, instead of further extending the context window size of current LLMs, we take a fresh perspective to understand and address this gap. Our core insight revolves around a phenomenon we term the *left-skewed position frequency distribution* – a pattern of severe undertraining of long-distance position indices during pretraining and post-training stages. This skewed distribution significantly contributes to the model's suboptimal performance in long-range modeling tasks. In SlimPajama-627B (Cerebras, 2023), a widely used pretraining corpus (Geng & Liu, 2023; Zhang et al., 2024b), we clearly observe this left-skewed phenomenon. As illustrated in Figure 1a, even with presumably adequate long-sequence data, the frequency of position indices decreases dramatically as distances increase. For instance, when training a model with a 2048 context length on SlimPajama, the frequency of position indices used to model relationships between distant tokens (distances $\geq 1024$) is less than 20%, and for even longer distances ($\geq 1536$), it drops below 5%. Probing experiments conducted during pretraining reveal that the frequency of exposure to specific position indices has a crucial impact on the training context utilization. Capturing long-range dependencies is inherently more challenging (Zhu et al., 2023; Wu et al., 2024), and this challenge is exacerbated when the frequency of position indices allocated to gather distant information is exceedingly low, as observed in Figure 1. In other words, the difficulty in modeling long-term dependencies, coupled with the undertraining of the positions responsible for them, provides a compelling explanation for the discrepancy between the theoretical and practical context lengths in LLMs.

Building on these findings, we investigate whether well-trained positions can be leveraged to capture information from distant inputs during inference. To address this, we propose a training-free approach called S̲hif̲T̲ed R̲otray position embedd̲ING (STRING). This method eschews the use of positions at the tail of the frequency distribution during inference. Specifically, STRING shifts position indices from the main diagonal of the position matrix toward its bottom-left corner. This adjustment enables the model to represent long-range dependencies using frequently encountered position indices, effectively approximating the undertrained ones. STRING can be efficiently implemented using Flash Attention (Dao, 2023) by combining two key components: (1) sliding window attention (Beltagy et al., 2020; Ding et al., 2023; Xiao et al., 2023; 2024) around the diagonal, and (2) self-attention at the bottom-left corner using shifted position indices (Algorithm 1). This implementation incurs no additional computational costs and causes no obvious slowdowns during inference.

By strategically overwriting position indices in the upper range of the training length, we achieve substantial performance enhancements across seven open-source LLMs with context lengths ranging from 2K to 128K on the Needle-in-a-Haystack (4-needle) test, resulting in an average score increase of 18 points. STRING requires no additional training, enabling seamless scaling up with powerful large-scale models such as Llama3.1 70B (Llama Team, 2024) and Qwen2 72B (Bai et al., 2023). This integration not only establishes new state-of-the-art performance for open-source LLMs on long-context benchmarks RULER (Hsieh et al., 2024) and InfiniteBench (Zhang et al., 2024d) but also enables Llama3.1 to outperform leading commercial models, including `GPT-4-128K` (OpenAI, 2023), Claude-2 (Anthropic, 2023), and Kimi-chat (Moonshot AI, 2023), across a wide range of synthetic and practical tasks. The substantial improvements achieved by STRING provide strong evidence for our hypothesis: underrepresented position indices at the tail of the position frequency distribution, strongly constrain the long-context capabilities of current LLMs. We hope our findings will inspire new approaches to overcome these limitations and lead to more effective long-context processing in future LLM designs.

## 2 LEFT-SKEWED POSITION FREQUENCY DISTRIBUTION

### 2.1 POSITION EMBEDDINGS IN LLMS

Self-attention mechanisms (Vaswani et al., 2017; Radford et al., 2018; Dai et al., 2019) inherently lack positional information (Liu et al., 2021; Su et al., 2022; Sun et al., 2022). To introduce positional information, a common approach is to design a function $p$. For an input at position $i$, we inject positional information using the following method: $\mathbf{h}_i = p(\mathbf{h}, i)$ where $\mathbf{h}$ is the hidden representation

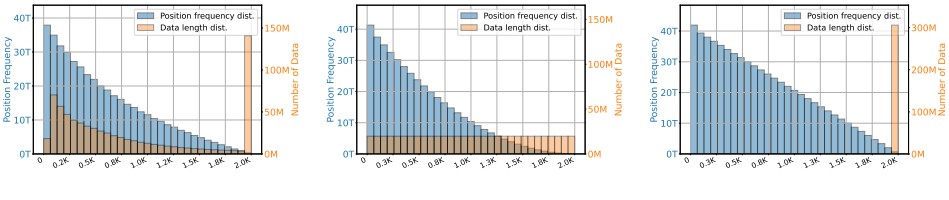

(a) Natural data distribution    (b) Uniform data distribution    (c) Concatenated data distribution

Figure 1: Position frequency distribution exhibits a pronounced left-skewed pattern across training data of varying lengths. Figure 1a illustrates the natural data length distribution of SlimPajama-627B where oversized data is truncated into multiple 2K sequences. Figure 1b presents the case with a uniform length distribution and the position frequency decline quadratically. Figure 1c demonstrates that when all data are concatenated into a 2K sequence, the position frequency decreases linearly with increasing position indices. The X-axis represents data length (shown in orange) and position indices (shown in blue). The left Y-axis indicates the frequency of each position, while the right Y-axis represents the number of data for each length.

of the input token. Another approach involves relative positional encodings (Bao et al., 2020), such as T5-bias (Raffel et al., 2023) and ALiBi (Press et al., 2022), which injects relative positional information by incorporating the relative distance $(i - j)$ when computing the attention score between the $j$-th token and the $i$-th token.

To achieve better training stability and lower perplexity, mainstream large models like Qwen (Hui et al., 2024) and Llama (Llama Team, 2024) employ Rotary Position Embedding (RoPE) (Su et al., 2022) as their positional encoding method. RoPE directly injects positional information into the query and key vectors, enabling the inner product to encode the relative position information between the query and key. We adopt the notation $p$ for the embedding function of RoPE. Considering the $i$-th query and the $j$-th key, we have: $\mathbf{q}_i = p(\mathbf{q}, i)$ and $\mathbf{k}_j = p(\mathbf{k}, j)$. When computing attention, the inner product $\mathbf{q}_i^\top \mathbf{k}_j$ contains only the relative positional information $(i - j)$, which means for any pair $(m, n)$ such that $m - n = i - j$, it holds that $\mathbf{q}_m^\top \mathbf{k}_n = \mathbf{q}_i^\top \mathbf{k}_j$.

## 2.2 RELATIVE POSITION MATRIX AND POSITION FREQUENCY

Using relative positional encodings implies that, given training length $L$, the resulting relative position matrix $P$ after computing $\mathbf{Q}^\top \mathbf{K}$ is defined by:

$$P = \begin{pmatrix} 0 & & & & \\ 1 & 0 & & & \\ \ddots & \ddots & \ddots & & \\ L-2 & \cdots & 1 & 0 & \\ L-1 & L-2 & \cdots & 1 & 0 \end{pmatrix} \tag{1}$$

where the Toeplitz matrix $P$ captures the relative positional relationships between tokens, with each element $P[m][n] = m - n$ encoding the relative distance between the $m$-th and $n$-th tokens in a sequence. Based on Eq. 1, we define the frequency of relative position $i$ by $f(i) = L - i$, which is the number of occurrences of a relative position $i$. Throughout the remainder of this paper, the term "position" refers to **relative position**. The structure of matrix $P$ is linearly skewed toward smaller positions, which inherently favors performance on shorter sequences. For example, when using a training context window of $L = 2048$ tokens, the relative position 2047 occurs only once in $P$.

The frequency of relative positions in $P$ also depends on the data length distribution of the pretraining corpus $\mathcal{C}$. We can obtain the frequency of relative position $i$ by the following equation:

$$f(i) = \sum_{s \in \mathcal{C}} \max(|s| - i, 0), \quad 0 \leq i < L \tag{2}$$

We observe that the position frequency distribution is usually highly *left-skewed*, indicating that the model is frequently exposed to small positions, while larger positions account for only a small proportion. To illustrate this phenomenon, we examine the position distribution when using SlimPajama-627B (Cerebras, 2023) as the training corpus. The blue bars in Figure 1 illustrate the position

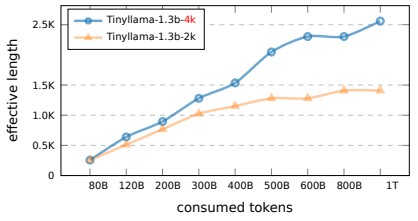
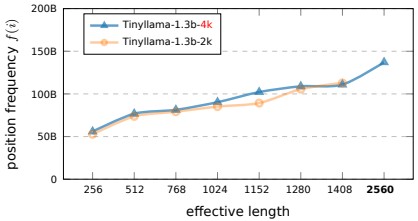

(a) Effective length vs. consumed tokens      (b) Effective length vs. position frequency

Figure 2: Analyzing effective context length of LLMs pretrained on SlimPajama with respect to training length, token consumption, and position frequency. In Figure 2b, we use the model effective length as the X-axis, and the Y-axis indicates the number of times the model was exposed to that specific position during training.

frequency distribution based on the natural data length distribution of SlimPajama. Specially, when the training length is 2048, the position indices $i \leq 1024$ account for more than 80% of all indices, whereas those with $i \geq 1536$ constitute less than **5%**. In addition to the biased relative position matrix $P$, the real-world data length distribution is also biased. Given a training context length of 2048 tokens, the data length distribution is shown in Figure 1a (orange bars): about 20% of the data consists of sequences around 256-512 tokens, and approximately 20% of the samples are around 2048 tokens. This latter percentage arises because long sequences are segmented into multiple sequences of length 2048, following popular open-source pretraining projects (Geng & Liu, 2023; Zhang et al., 2024b). Due to the combined effect of the data distribution and the relative position matrix, the frequency of positions decreases following a polynomial trend as the position indices increase.

Despite capturing local dependencies is often effective for LLMs, the imbalance in position frequency distribution when modeling both local and long-range dependencies is more pronounced than expected. This may result in a substantial underrepresentation long-range dependencies.

## 3   A PROBING EXPERIMENT ON POSITION FREQUENCY AND MODEL EFFECTIVE LENGTH

In this section, we empirically investigate the impact of the left-skewed position frequency distribution on the effective context length of LLMs. Since the training data distributions for most open-source LLMs are opaque and cannot be directly analyzed by researchers, this study represents the first exploration of the impact of position frequency during the pretraining stage.

**Evaluation**   To measure the effective context length, we adopt the popular Needle-in-a-Haystack task (gkamradt, 2023). We use the 4-needle setting, the same as described in the Llama 3.1 report (Llama Team, 2024), which involves inserting four needles (6-digit numbers (Hsieh et al., 2024; Mohtashami & Jaggi, 2023)) into the context at various positions. The model should perfectly retrieve at least two of them. The input examples used in this experiment can be found in Table 5 of the Appendix. The evaluation context length increases in 128-token steps until the model fails to correctly find 2 of 4 inserted needles. We perform 500 tests at each length.

**Experimental Setup**   We pretrain two 1.3B-parameter models (referred to as TinyLlama-1.3B) from scratch on the natural data distribution of the SlimPajama dataset to observe changes in the model's effective length. The total training tokens are 1T and we evaluate the model's effective context length for every 10B tokens during training. Both models begin to exhibit needle-retrieval ability after about 50B tokens of training. Since position frequency is difficult to control directly, we perform controlled experiments by adjusting two factors: (1) consumed tokens, and (2) the training context window size. The first factor is straightforward. For the second factor, we illustrate the position frequency distribution after training with 1T tokens using different training lengths (2K and 4K) in Figure 3. The configuration of our pretraining codebase and models is detailed in Section A.2.

**Findings** Following previous work (Kaplan et al., 2020), we demonstrate how the models' effective length grows with increasing training tokens for two different training lengths (*Finding* (1)), while our further analysis reveals that the position frequency is the underlying factor (*Findings* (2) and (3)).

*(1) Larger training context window consumes fewer tokens to achieve the same effective context length*: In Figure 2a, a notable observation is that training with longer sequences results in a greater effective context length when the same number of tokens is consumed. Specifically, the model trained with a sequence length of 4K tokens achieves an effective context length of 1.4K after consuming 400B tokens. In contrast, the model with a 2K training length needs around **1T** tokens to attain the same effective context length. But it does not mean larger context training window can save more computation.

*(2) Models can achieve similar effective context lengths if they have been exposed to similar frequencies of position indices, even if their maximum training lengths differ*: By directly plotting the effective context length against the frequency of position indices used to model that length (Figure 2b), we observe that the growth trends of effective lengths for different models align when the Y-axis represents the frequency of indices at that length. For instance, when the effective context length reaches 1,280 tokens, both models exhibit a position frequency $f(1280)$ of 100B. This indicates that models can attain comparable effective context lengths when they have been trained on similar frequencies of position indices, regardless of differences in their maximum training lengths.

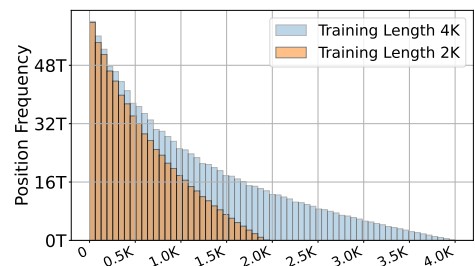

Figure 3: Position frequency distribution for models trained with different training lengths after consuming 1T tokens. With the same number of tokens, training length has little effect on small relative positions. For example, the relative position 0 appears 4K times in both a single 4K sequence and two 2K sequences with the same total token count of 4K in each case.

*(3) The growth trend of the model's effective length aligns with the position frequency distribution*: In Figure 3, we observe that models with different training lengths have close position frequencies when the position index $i \leq 1024$. As $i$ continues to increase, the frequency gap between models trained with 4K and 2K context lengths becomes increasingly larger. The growth rates of these two models' effective lengths also align with this trend (Figure 2). Both models consume roughly the same number of tokens (around 300B) when reaching an effective length of 1024. However, as the effective length increases further, the growth rate of the model pretrained with a 2K context window becomes significantly slower.

**Limitations in Gathering Distant Inputs** We visualize the performance of infrequent positions with the Needle-in-a-Haystack (4-needle) test (gkamradt, 2023). The distance between the query and the needles increases as the depth becomes smaller and the testing context length becomes longer. The results indicate that when the needle and query are far apart, both TinyLlama 1.3B and the latest Llama3.1 8B model fail to retrieve the needle effectively. In Figure 4, when we place the query at the end of the document, we find that models fail at retrieving information from the *beginning* of the document. Concretely, in Llama3.1, performance significantly degrades when position indices exceed 90K. TinyLlama struggles to gather information when the distance exceeds 1,536 tokens. We also evaluate 13 models from the open-source community, as shown in Table 4, and find that most failure cases occur within the first $\frac{L}{3}$ of the document. This may indicate that the last $\frac{L}{3}$ positions of current LLMs all fall in the tail of the position frequency distribution.

## 4 SHIFTED ROTARY POSITION EMBEDDING

In Figure 1c, we demonstrate that even when all data are concatenated to fill the training context window, positions at the tail remain infrequent. In this section, we introduce Shif̲T̲ed R̲otray position embedd̲IN̲G (STRING), STRING shifts position indices from the diagonal of $P$ towards its bottom-left corner, allowing the model to gather distant information with frequent position indices.

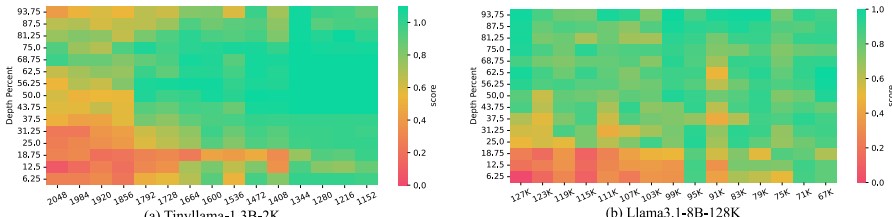

Figure 4: NIAH results for our pretrained model TinyLlama-1.3B (2K) and Llama3.1 (128K) where the X-axis means input context length and the Y-axis represents the document depth. In this figure, we clearly observe that for TinyLlama 2K and Llama3.1 128K, most poor-performing cases are concentrated in the lower-left triangle, indicating that the models are unable to gather distant needles.

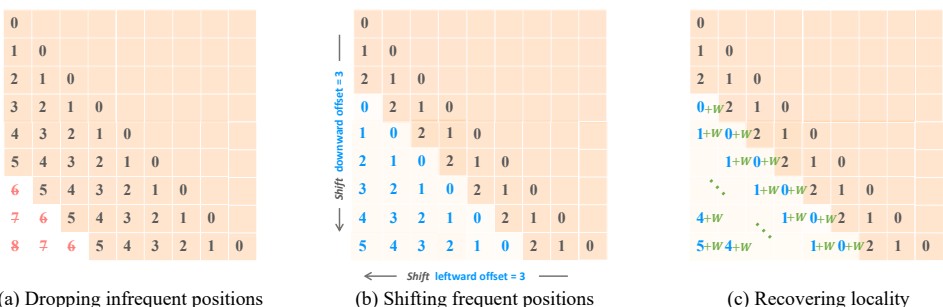

Figure 5: A illustrative example of STRING for a sequence length of $L = 9$. (a) Position indices 6, 7, and 8 are removed from the matrix. (b) Indices 0, 1, 2, 3, 4, and 5 are shifted from the main diagonal to the lower-left triangle with an offset of 3. (c) A small constant $W$ is added to all diagonals where $m \geq n - 3$, thereby restoring emphasis on the neighboring $W$ tokens. The position matrix of Llama3.1-128K using STRING is shown in Figure 8 Appendix.

## 4.1 MANIPULATING THE POSITION MATRIX

STRING is implemented by manipulating the position matrix $P$. The three main procedure of STRING is shown in Figure 5:

*(1) Dropping Infrequent Positions*: We begin by assuming that position indices greater than a threshold $N$ falls into the infrequent area. Consequently, STRING initially drops all position indices $i \geq N$. As depicted in Figure 5a, we set $N = 6$ and $L = 9$, resulting in the removal of position indices 6, 7, and 8 from the matrix and leaving an empty area.

*(2) Shifting Frequent Positions*: Next, we shift the remaining position indices from the main diagonal (the high-frequency area) to fill the empty triangle in the bottom-left corner of $P$. The shift offset is defined as $S = L - N$. In our example, $S = 9 - 6 = 3$, as shown in Figure 5b. For instance, let's consider the last row of the matrix $P$. The position indices after dropping are $[-, -, -, 5, 4, 3, 2, 1, 0]$. To fill the 3 empty slots, we shift the positions leftwards with a stride of 3, and they become $[5, 4, 3, 2, 1, 0, 2, 1, 0]$. Formally, the updated position matrix is defined as:

$$P[m][n] = \begin{cases} P[m][n] - S & \text{if } m \geq n - S, \\ P[m][n] & \text{otherwise.} \end{cases} \quad (3)$$

Here, $m, n$ is the row/column index, $m = n - S$ indicates that the element is located on a diagonal of $S$ away from the main diagonal, and $m \geq n - S$ signifies that the element is in the lower-left region relative to this diagonal. The resulting position matrix after this operation is shown in Figure 5b.

*(3) Restoring Locality with a Small Window*: Applying Eq. 3 disrupts the model's ability to capture local relationships because it alters the relative positions between neighboring tokens (Su, 2023; Jin et al., 2024; An et al., 2024a). Specifically, the relative positions on the $S$-th diagonal are set to zero. Since neighboring tokens are crucial for generating fluent content, we introduce a small local window value $W \ll S$ for elements where $m \geq n - S$, as illustrated in Figure 5c. This adjustment maintains

emphasis on the closest $W$ neighboring tokens. The final position matrix is defined as:

$$P[m][n] = \begin{cases} P[m][n] - S + W & \text{if } m \geq n - S, \\ P[m][n] & \text{otherwise.} \end{cases} \quad (4)$$

In Eq.4, $S$ is the shift offset, and $W$ is used to ensure the neighboring $W$ tokens remain the closest in terms of positional encoding. Notably, $W$ does not rely on $L$, whereas $S$ heavily depends on $L$. We suggest setting the local window $W \geq 32$ and the offset $\frac{L}{3} \leq S \leq \frac{L}{2}$. We set $S = \frac{L}{3}$ and $W = 128$ for all models across downstream tasks. An ablation study is shown in Figure 7.

**FlashAttention Implementation** We implement STRING using FlashAttention (Dao et al., 2022), which is essential for verifying the method on modern large language models (LLMs) that typically have long context windows (e.g., 128K tokens). STRING can be efficiently implemented by modifying the position indices used in RoPE and combining two attention patterns. The pseudocode for STRING is provided in Algorithm 1. Our implementation splits the standard self-attention mechanism into two components:

1. **Sliding Window Attention** (`lines 11-13`): This approach calculates the attention outputs around the main diagonal by considering positions where $m < n - S$ (`line 13`). When computing the sliding window attention, there is no need to modify the position indices for either queries (`line 6`) or keys (`line 7`).

2. **Shifted Self-Attention** (`lines 15-19`): This method computes the attention outputs in the bottom-left triangle, specifically for positions where $m \geq n - S$, utilizing causal self-attention (`line 19`). In this process, the position indices for queries are replaced with shifted position indices (`line 16`). STRING controls the relative distance by only modifying the position indices for queries and there is no influence on caching keys and values.

Finally, we merge the attention outputs from the sliding window around the main diagonal and the left-bottom triangle to produce the final output. An example of applying STRING on Llama3.1 is shown in Section §A.1 and the efficiency test of STRING is shown in Figure 9.

---

**Algorithm 1** Pseudocode of STRING with FlashAttention

```
1  # Q, K, V: tensors with shape [L, d]
2  # W:  the local window value (scalar)
3  # S:  the slding window size (scalar)
4  # N:  the left-bottom triangle height (scalar)
5
6  pids_query = [0,1,2,...L-1] # standard position ids for keys
7  pids_key = [0,1,2,...L-1] # standard position ids for queries
8  # Apply rotary position embeddings to K
9  K = apply_rotary_pos_emb(K, pids_key)
10
11 # <--- Calculating sliding window attention around the diagonal --->
12 Q_diag = apply_rotary_pos_emb(Q, pids_query)
13 O_diag, attn_map_diag = flash_attn(Q_diag, K, V, sliding window=S)
14
15 # <--- Calculating self-attention at the left-bottom triangle --->
16 pids_q_shifted = pids_query - S + W # new position ids for queries
17 Q_shifted = apply_rotary_pos_emb(Q, pids_q_shifted)
18 # obtain q,k,v in the bottom-left corner & calculate flash-attn
19 O_shifted, attn_map_shifted = flash_attn(Q_shifted[-N:], K[:N], V[:N])
20
21 # Merge the attention outputs from the diagonal and left-bottom triangle
22 output = merge_diag_shifted(O_diag, O_shifted, attn_map_diag, attn_map_shifted)
```

---

Figure 6: Detailed pseudocode of STRING incorporating FlashAttention Dao et al. (2022). The implementation of `merge_diag_shifted` can be found in Algorithm 2 in the Appendix.

## 4.2  MAIN RESULTS OF STRING

In this section, we evaluate the effectiveness of STRING across three widely recognized long-context benchmarks: Needle-in-a-Haystack (NIAH) (gkamradt, 2023), RULER (Hsieh et al., 2024), and InfiniteBench (Zhang et al., 2024d). These tasks enable us to assess STRING's performance across a broad spectrum of practical scenarios.

**Baselines** We primarily compare STRING with the original position embedding RoPE used in mainstream Large Language Models. Additionally, we evaluate RoPE against several effective extrapolation baselines. Specifically, we compare STRING with the following training-free extrapolation methods: NTK-Aware RoPE (LocalLLaMA, 2023b;a), YaRN (Peng et al., 2023), ReRoPE (Su, 2023), Self-Extend (Jin et al., 2024), and DCA (An et al., 2024a). Extrapolation refers to testing LLMs on sequence lengths beyond their training lengths while STRING focus on improving the performance within the training context size. NTK-Aware RoPE and YaRN implement extrapolation by increasing the base frequency of RoPE. Meanwhile, ReRoPE, Self-Extend, and DCA modify the position matrix to aviod unseen positions. We reproduced their results using scripts from their official repositories. When testing these extrapolation baselines, we modify the training length of the model to $\frac{2}{3}$ of the original length and set the extrapolation scaling factor to $\frac{L_{\text{test}}}{L_{\text{train}}} = \frac{3}{2}$, meaning the test sequence length is 1.5 times the training length. All other configurations remain the same as in their paper. Our findings indicate that although extrapolation methods can extend the model's capability to handle longer sequences, the performance improvements are still limited within the original training length.

Table 1: Needle-in-a-haystack (4 needles) results of 7 base models across various methods (columns reordered from smallest to largest average) where $L_{train}$ means the size of the training context window. All the models were tested using their training length. The number of test cases is 500.

| Model | $L_{train}$ | ReRoPE | NTK | RoPE(origin) | Self-Extend | YaRN | DCA | STRING |
|---|---|---|---|---|---|---|---|---|
| TinyLlama-1.3B (ours) | 2k | 62.8 | 62.0 | 56.6 | 60.2 | 68.6 | 74.4 | **84.6** |
| TinyLlama-1.1B-3T | 2k | 77.2 | 79.8 | 69.8 | 83.2 | 88.0 | 80.2 | **97.2** |
| Llama-2-7B | 4k | 98.6 | 98.6 | 98.0 | 95.4 | 98.0 | 91.6 | **100.0** |
| Llama-3-8B | 8k | 99.6 | 100.0 | 99.8 | 99.8 | 100.0 | 99.9 | 99.6 |
| LWM-7B-base | 32k | 25.2 | 19.4 | 31.8 | 29.0 | 22.2 | 28.8 | **50.4** |
| Mistral-7B-base | 32k | 54.5 | 42.2 | 52.8 | 54.2 | 48.2 | 64.2 | **73.0** |
| Llama-3.1-8B | 128k | 53.6 | 71.2 | 66.0 | 65.8 | 68.8 | 72.8 | **95.2** |
| **Average** | – | 67.3 | 67.6 | 67.8 | 69.6 | 70.5 | 73.1 | **85.7** |

**Needle-in-a-Haystack** Needle-in-a-Haystack (gkamradt, 2023) (NIAH) is the most popular long-context task, extensively utilized in recent studies (Zheng et al., 2024; Liu et al., 2024b). As reported by Hsieh et al. (2024); Wang et al. (2024a), single needle retrieval is no longer a challenging task for current LLMs, and we adopt the multi-needle setting following Llama 3.1 (Llama Team, 2024) and the input example can be found in Table 5. We verify the effectiveness of our method on seven community models with training lengths ranging from 2K to 128K. Across all seven models, LargeWorldModel (LWM-7B-base) (Liu et al., 2024a), Mistral 7B (Mistral.AI, 2024), and Llama 3.1 8B (Llama Team, 2024) are continually trained on longer contexts. On models with various training context lengths, STRING consistently outperforms other methods, achieving the highest scores on each model. Notably, STRING improves the average performance by a significant margin, reaching 85.7% compared to the next best method, DCA, at 73.1%, and the original RoPE at only 67.8%.

**RULER** The RULER benchmark (Hsieh et al., 2024) encompasses a variety of synthetic tasks, including eight variants of Needle-in-a-Haystack (NIAH), as well as tasks involving variable tracking, counting, and long-context question answering (QA). The evaluation code and metrics are from their official repository[1]. The primary results are presented in Table 2. The results on Llama3.1-8B reveal that, except for our proposed method (STRING), all other extrapolation-based approaches fail to achieve performance improvements. Since our method does not require additional training, we are able to validate its effectiveness on 70B-level models. Applying our method to larger models yields remarkable enhancements: a *15-point improvement* on Llama3.1 70B and over a *30-point improvement* on Qwen2 72B compared to the baseline. Furthermore, our approach achieved state-of-the-art performance on the RULER benchmark for open-source models. Notably, after applying STRING, both Llama3.1 70B and Qwen2 72B surpass `GPT-4-128K` in average performance. The remarkable performance gain on large models demonstrates that the frequent positions in large models may possess a stronger potential for modeling long-range dependencies. Additionally, we also demonstrate that both Llama3.1 and Qwen2 can be effectively boosted to an effective sequence length of 100K on RULER by STRING (the last block in Table 2).

---

[1]https://github.com/hsiehjackson/RULER

Table 2: Performance of various models and methods on RULER with a tested at a sequence length of 128K. The RULER benchmark consists of 13 tasks (500 test cases for each task) categorized into Needle-in-a-Haystack (NIAH), Variable Tracing (VT), Aggregation, and Question Answering (QA). We report the average scores for each category as well as the overall average across all 13 tasks. **Effective** denotes the actual effective sequence length as defined in RULER, indicating whether the model surpasses the performance of Llama2 (Touvron et al., 2023b), and **Claimed** represents the sequence length reported by the model.

| Models | Effective/Claimed | NIAH | VT | Aggregation | QA | Avg. (13 tasks) |
|---|---|---|---|---|---|---|
| Llama2-chat | 4K / 4K | 96.9 | 89.7 | 84.8 | 49.7 | 85.6 |
| GPT-4-1106-preview | 64K / 128K | 84.8 | 99.6 | 79.7 | 59.0 | 81.2 |
| GLM4 (*Open-source best*) | 64K / 1M | 94.4 | 97.7 | 49.7 | 63.6 | 83.1 |
| LWM (7B) | 4K / 128K | 83.4 | 15.2 | 29.1 | 52.6 | 65.0 |
| Phi3-medium (14B) | 8K / 128K | 51.3 | 26.0 | 43.5 | 38.0 | 46.1 |
| Llama3.1 (8B) | 32K / 128K | 92.6 | 70.4 | 36.2 | 58.8 | 77.0 |
| + YaRN | 32K / 128K | 94.7 | 39.8 | 38.2 | 58.8 | 76.3 |
| + DCA | 32K / 128K | 89.5 | 62.5 | 39.2 | 55.2 | 74.4 |
| + Self-Extend | 32K / 128K | **94.9** | 65.0 | 37.3 | 49.8 | 76.8 |
| + ReRoPE | 32K / 128K | 90.0 | 56.3 | 38.7 | 56.9 | 74.4 |
| + STRING | 32K / 128K | 94.0 | 88.1 | 37.6 | 62.7 | 80.0 |
| Yi (34B) | 32K / 200K | 90.2 | 76.8 | 43.4 | 59.9 | 77.3 |
| GradientAI/Llama3 (70B) | 16K / 1M | 84.9 | 56.2 | 41.4 | 59.8 | 72.1 |
| Mixtral (8x22B) | 32K / 64K | 23.8 | 0.0 | 69.7 | 40.8 | 31.7 |
| Command-R-plus (104B) | 32K / 128K | 65.7 | 97.2 | 59.5 | 39.2 | 63.1 |
| Llama3.1 (70B) | 64K / 128K | 78.9 | 59.2 | 39.8 | 47.6 | 66.6 |
| + STRING | 100K / 128K | 92.7 | 95.6 | 50.0 | **63.0** | 81.7 |
| Qwen2 (72B) | 64K / 128K | 48.0 | 79.0 | 70.3 | 47.2 | 53.7 |
| + STRING (*new SOTA*) | 100K / 128K | 91.2 | **98.4** | **83.7** | 52.2 | **84.6** |
| **Test Length:** *100K* | | | | | | |
| Llama3.1-STRING (70B) | 100K / 128K | 94.6 | 97.8 | 72.1 | 67.3 | 87.2 |
| Qwen2-STRING (72B) | 100K / 128K | 93.9 | 97.7 | 88.1 | 57.8 | 87.8 |

Table 3: Comparison of STRING with three leading commercial long-context models on InfiniteBench. Each model is evaluated using a maximum context length of 128K.

| Tasks | Commercial Models | | | Llama3.1 8B | | Llama3.1 70B | |
|---|---|---|---|---|---|---|---|
| | GPT-4 | Claude2 | Kimi-chat | RoPE(origin) | STRING | RoPE(origin) | STRING |
| En.Sum | 14.73 | 14.45 | 17.93 | 26.00 | **28.22** | 26.89 | 27.64 |
| En.QA | **22.22** | 11.97 | 16.52 | 10.05 | 10.20 | 13.68 | 16.73 |
| En.MC | 67.25 | 62.88 | 72.49 | 65.50 | 70.30 | 76.41 | **81.98** |
| En.Dia | 8.50 | **46.50** | 11.50 | 20.00 | 19.50 | 18.00 | 30.50 |
| Retr.PassKey | 100.00 | 97.80 | 98.14 | 100.00 | 100.00 | 100.00 | **100.00** |
| Retr.Number | 100.00 | 98.14 | 94.42 | 99.32 | 99.89 | 100.00 | **100.00** |
| Retr.KV | **89.00** | 65.40 | 53.60 | 42.00 | 83.00 | 2.22 | 76.07 |
| Code.debug | **39.59** | 2.28 | 18.02 | 22.84 | 26.90 | 29.20 | 32.80 |
| Math.find | **60.00** | 32.29 | 12.57 | 32.18 | 34.87 | 40.92 | 46.28 |
| **Avg.** | 55.69 | 47.96 | 43.91 | 46.43 | 52.54 | 45.25 | **56.88** |

**InfiniteBench** InfiniteBench (Zhang et al., 2024d) encompasses a variety of real-world tasks, including long-context question answering (QA), multiple-choice QA, mathematical problem-solving, long-dialogue QA, long-context summarization, retrieval tasks, and code debugging.

The evaluation code and metrics are sourced from the official repository[2]. The results for commercial models are from Zhang et al. (2024d). We compare our method, STRING, with the original position

---
[2] https://github.com/OpenBMB/InfiniteBench

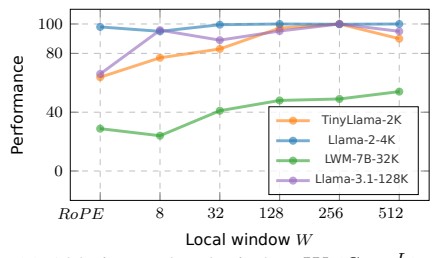 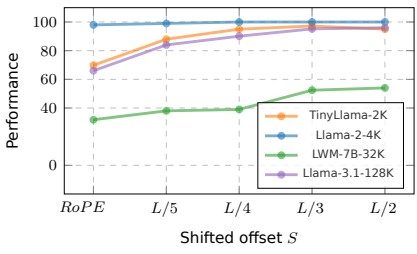

(a) Ablation on local window $W$ ($S = \frac{L}{3}$)  (b) Ablation on shifted offset $S$ ($W = 128$)

Figure 7: Ablation study on the local window $W$ and shifted offset $S$ where $L$ is the training length.

embedding, RoPE, across two scales of Llama3.1: 8B and 70B parameters. The results are presented in Table 3. STRING demonstrates significant improvements for both models; for instance, we enhance the performance of Llama3.1 70B by over 10 points, establishing a new state-of-the-art for open-source models. On InfiniteBench, our method also surpasses the performance of strong baseline GPT-4-128K and significantly outperforms Claude-2 and Kimi-chat.

**Ablation Study**  We conduct an ablation study on the Needle-in-a-Haystack (4 needles) task to examine the impact of two main hyperparameters in our STRING: the local window size $W$ and the shifted offset size $S$. The experimental results are shown in Figure 7. We increase the local window size from 4 to 512 and find that when $W \geq 32$, the model achieves a significant improvement compared to the original RoPE method. Furthermore, as long as $W \ll S$, further increasing $W$ does not cause a performance drop. For the offset size $S$, we experiment with values ranging from $\frac{L}{5}$ to $\frac{L}{2}$. As $S$ increases, more position indices are discarded. We observe that within this range, the performance increased with the growth of $S$. However, the trend slowed down when $S$ exceeded $\frac{L}{3}$, indicating that at least the last 33% to 50% of the position can be overwritted.

## 5  RELATED WORK

**Long-Context Scaling of LLMs** Modeling long text has always been a challenging problem. With the development of large language models (LLMs), researchers have begun to explore ways to extend these models to handle longer contexts from various perspectives. (1) Efficient Architectures: Jiang et al. (2024); Fu et al. (2024a); Ding et al. (2023); Song et al. (2023); Yang et al. (2024); Zhu et al. (2024b) demonstrate that the training and inference overhead of long-context LLMs can be substantially optimized by sparse attention patterns. Another crucial architecture is state space models (Gu & Dao, 2023; Yuan et al., 2024; Lieber et al., 2024). (2) Continual Training with Long Data: Efforts have been made to continually train models by collecting high-quality long sequences (Fu et al., 2024b; Zhu et al., 2024a; Wu et al., 2024; Gao et al., 2024). (3) LLMs with Infinite Contexts: Recent work has shown that the context length of LLMs can be scaled to infinite, as evidenced by models such as StreamingLLM and InfLLM (Xiao et al., 2023; 2024; Han et al., 2023; Zhang et al., 2024a; Cai et al., 2024; Lin et al., 2024a; Dong et al., 2024). However, these methods typically cannot maintain a full KV cache, resulting in weakened long-context capabilities.

**Length Extrapolation** Training to extend the model context length incurs significant overhead. Recent works focus on length extrapolation, training on short sequences to infer longer ones, as a means to address this issue (Press et al., 2022; Raffel et al., 2023; Han et al., 2024). An et al. (2024a); Jin et al. (2024); Su (2023); Ma et al. (2024); Zhang et al. (2024e) believe that the model's inability to generalize to longer contexts is caused by positions being out-of-distribution. They achieved effective extrapolation by repeating trained positions, thereby maintaining low perplexity in exceedingly long contexts. On the other hand, Zhu et al. (2023) randomly places large position indices within the training window in the training and infer longer sequences. For RoPE-based LLMs, Peng et al. (2023); Men et al. (2024); Zhong et al. (2024); Wang et al. (2024b) reduce the long-range attenuation effect of RoPE by amplifying the base frequency, thereby bringing the remote token closer.

## 6 CONCLUSION

This work uncovers the limitations of current open-source large language models in effectively utilizing their extended training context windows. We show that using positions at the tail of the left-skewed position frequency distributions strongly hinders models' long-range dependency modeling ability. We introduce STRING, a novel approach that shifts well-trained positions to replace ineffective ones during inference, thereby enhancing the model's ability to capture distant contextual information without requiring additional training. Our experiments demonstrate that STRING significantly boosts the performance of strong baselines like Llama 3.1 70B and Qwen-2 72B on prominent long-context benchmarks, setting new state-of-the-art results for open-source LLMs.

## ACKNOWLEDGMENTS

We sincerely appreciate the assistance provided by various individuals and teams. The pretraining part of this work is built upon the TinyLlama Project. We would also like to express our gratitude to the Llama3 and Qwen2 teams for providing the robust base models that have been essential for this work. Additionally, we thank Yushi Bai for his help and valuable discussions. This research was supported in part by the joint research scheme of the National Natural Science Foundation of China (NSFC) and the Research Grants Council (RGC) under grant number N_HKU714/21.

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

# A APPENDIX

## A.1 APPLYING STRING ON LLAMA3.1 128K

In this section, we demonstrate the application of STRING on Llama3.1 128K. We present the utilization of STRING to drop position indices greater than $\frac{2}{3} * L \approx 42\text{K}$ and $\frac{1}{2} * L = 64\text{K}$, where $L$=128K represents the training length of Llama3.1. The resulting position matrix is illustrated in Figure 8. In Figure 8a, let us consider the last row of the matrix. The original position indices are $[128\text{K} - 1, \ldots, 2, 1, 0]$. After dropping position indices $\geq 86\text{K}$, they become $[\underbrace{-, -, \ldots, -}_{\text{42K empty slots}}, \underbrace{86\text{K} - 1, \ldots, 2, 1, 0}_{\text{86K indices}}]$. To fill the empty slots, we shift the positions leftwards with a stride of $S = 42\text{K}$, resulting in $[86\text{K} - 1, \ldots, 2, 1, 0, 42\text{K} - 1, \ldots, 2, 1, 0]$. After adding a local window $W$ of 128, we obtain the shifted position indices: $[86\text{K} + 127, .., 129, 128, 42\text{K} - 1, \ldots, 2, 1, 0]$. Applying STRING with an offset $S = 64\text{K}$ is shown in (Figure 8b). The procedure is the same. We also illustrate the changes in the last row of the position matrix. After dropping position indices $\geq 64\text{K}$, the row is converted to $[\underbrace{-, - \ldots, -}_{\text{64K empty slots}}, 64\text{K} - 1, \ldots, 2, 1, 0]$. Then, the well-trained positions are shifted from the diagonal: $[\underbrace{-, - \ldots, -}_{\text{64K empty slots}}, 64\text{K} - 1, \ldots, 2, 1, 0] \rightarrow [64\text{K} - 1, .., 1, 0, 64\text{K} - 1, \ldots, 1, 0]$. Finally, the position indices after adding a local window of 128 are $[64\text{K} + 127, .., 129, 128, 64\text{K} - 1, \ldots, 1, 0]$.

**Projecting Local Relationships** The local value $W$ is essential for projecting the relationships among neighboring tokens, ensuring fluent generation. To demonstrate the importance of setting local value $W$, consider the relative matrix of Llama3.1 128K, where the value on the 42K-th diagonal is set to $W$ instead of 42K (using ROPE) to achieve a shifting effect. This means each token treats the token 42K positions away as if it's at a distance of $W$. In an extreme case, if $W = 0$, the current token would treat the token 42K positions away as itself. As we know, LLMs rely heavily on neighboring tokens to maintain fluency. If $W$ is too small, the model can not correctly focus on these nearby tokens. By setting $W = 128$, we ensure that relative positions 0-127 are unique and assigned to neighboring tokens. This maintains fluency while enhancing long-context performance.

**Handling Short-Context Input** STRING will produce the same results as RoPE on short-context benchmarks. When the input length $l < S$ where $S$ is the shifting offset, the sliding window attention which calculates attention within in neighboring $S$ tokens. For Llama3.1 we have $S$=42K which significantly exceeds the input length of short context tasks. The sliding window attention in line 13 of Algorithm 6 calculates full attention for the short sequence without shifting positions, and the shifted self-attention mechanism is not invoked.

(a) Shifted offset $S = \frac{L}{3}$   (b) Shifted offset $S = \frac{L}{2}$

Figure 8: The resulted position matrix of Llama3.1 128K after shifting. In Figure (a), we use a shifted offset of $\frac{L}{3} \approx 42\text{K}$ and the local window $W$ is 128. In Figure (b), we overwrite more infrequent positions and the shifted offset is $S = \frac{L}{2} = 64\text{K}$.

## A.2    Pretraining Setup

We pretrain two 1.3B models with maximum context window sizes of 2048 and 4096 to observe how the models gain the effective context length. The model architecture aligns with TinyLlama 1.1B[3]. We utilize a hidden size of 2,048, the size of the feed-forward layers inside each transformer block is set to 5632. The model employs 32 attention heads and comprises 22 layers. The only difference is the use of the llama3 tokenizer (Llama Team, 2024), which has a larger vocabulary size of 128,256 tokens compared to the 32,000 tokens in TinyLlama 1.1B. This difference results in a larger embedding matrix. We used the SlimPajama-627B (Cerebras, 2023) dataset as our pretraining corpus and total training tokens for each model is 1T tokens.

Our pretraining codebase is primarily built on the TinyLlama project[4], a popular codebase for reproducing Llama at the 1B scale. The main speed optimization libraries employed in this project are Fully Sharded Data Parallel (FSDP)[5], FlashAttention-2 (Dao, 2023)[6], and xFormers (Lefaudeux et al., 2022)[7]. The entire project is based on PyTorch Lightning [8]. We use the cross entropy loss as the pretraining objective and the AdamW optimizer (Loshchilov & Hutter, 2019). Additionally, we employed a cosine learning rate schedule with a maximum learning rate of $4 * 10^{-4}$, starting from a minimum learning rate of $4 * 10^{-5}$. The warmup steps are 2,000. The batch size is set to 4M tokens for different training context lengths. For the model pretrained with a 4K context length, the gradient accumulation is set to twice that of the model trained with a 2K context length. We pack the sequences in a mini-batch into a long sequence and used the variable-length version of Flash Attention[9] to calculate casual self-attention on packed sequences. A gradient clipping threshold of 1.0 is used to stablize the gradient.

We utilized 16 NVIDIA 80G A100 GPUs on 2 nodes. Training a 1.3B model with a 2K context length and 1T tokens took approximately 28 days, while expanding the context length to a 4K context length took around 32 days.

## A.3    Efficiency Test of STRING

In this section, we demonstrate that STRING can be implemented with negligible additional overhead compared to flash attention by comparing the inference time and GPU memory consumption. We test the baseline and STRING on a single NVIDIA 80G A100 GPU based on Llama3.1 8B. The long inputs are sourced from the summarization task in InfiniteBench (Zhang et al., 2024d). We test the model 50 times and report the average results. The results of inference time are shown in Figure 9a, where we test the model with context lengths ranging from 64K to 128K. STRING maintains the average time consumed per token within 0.3 seconds of the standard Flash Attention. Figure 9b shows the consumption of GPU memory, with the growth of input context lengths, STRING exhibiting only a less than 5GB increase.

## A.4    Limitations

One limitation of this work is that it only investigates pretraining lengths smaller than 4K tokens, while the question of how to effectively implement long-context training remains an open open. The open-source community's approaches to this problem remains diverse (Hu et al., 2024; Fu et al., 2024b; An et al., 2024a; Jin et al., 2024). For companies, Llama3.1 (Llama Team, 2024) reported using a 6-stage training approach to gradually implement long-context training, but this makes it difficult to analyze position frequencies because the data distribution used in each stage is unknown.

STRING achieves surprising results by only using frequent position during inference. It is clear that there are many ways to adjust the distribution of frequent positions during training, but this may

---

[3] https://huggingface.co/TinyLlama/TinyLlama-1.1B-intermediate-step-1431k-3T/blob/main/config.json

[4] https://github.com/jzhang38/TinyLlama

[5] https://huggingface.co/docs/accelerate/usage_guides/fsdp

[6] https://github.com/Dao-AILab/flash-attention

[7] https://github.com/facebookresearch/xformers

[8] https://github.com/Lightning-AI/pytorch-lightning

[9] https://github.com/Dao-AILab/flash-attention/blob/main/flash_attn/flash_attn_interface.py#L1178

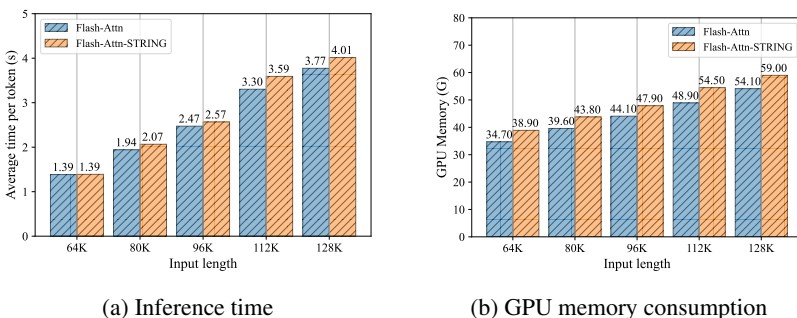

(a) Inference time  (b) GPU memory consumption

Figure 9: Efficiency Test of STRING and the standard Flash Attention based on Llama3.1 8B. All experiments are run on a single NVIDIA 80G A100 GPU.

require data with a distribution similar to the Llama training corpus to avoid the model losing its reasoning ability. A key feature of STRING is that it can be easily applied to all existing models without requiring the collection of high-quality data for training. We leave the problem of addressing the left-skewed distribution from a training perspective as a future work.

---

**Algorithm 2** Pseudocode of `merge_diag_shifted`

---

```
1  def merge_diag_shifted(O_diag, O_shifted, attn_map_diag, attn_map_shifted):
2      """
3      Merge the attention outputs from the diagonal and left-bottom triangle.
4
5      Parameters:
6      O_diag (Tensor: [L, d] ): Output tensor from diagonal attention.
7      O_shifted (Tensor: [N, d]): Output tensor from left-bottom triangle attention.
8      attn_map_diag (Tensor: [L, L]): Attention map from diagonal attention.
9      attn_map_shifted (Tensor: [N, N]): Attention map from left-bottom triangle attention.
10
11     Returns:
12         output (Tensor: [L, d] ): Merged output tensor.
13     """
14
15     #  the softmax normalizer of the sliding window attention
16     S=L-N # S is the slding window size, and N is the triangle height
17     diag_norm = attn_map_diag.sum(-1) # shape: [L,]
18     #  the softmax normalizer of the self-attention
19     shifted_norm = attn_map_shifted.sum(-1) # shape: [N,]
20     O_diag_head = O_diag[:S] # shape: [S, d], no need for changing the first S tokens
21     O_diag_tail = O_diag[S:] # [N, d]
22     diag_norm_tail = diag_lse[S:] #  [N,]
23     diag_rate = diag_norm_tail / (diag_norm_tail + shifted_norm) # [N,]
24     shifted_rate = shifted_norm / (diag_norm_tail + shifted_norm) # [N,]
25     O_merged_tail = diag_rate * O_diag_trail + shifted_rate * O_shifted  # [N,d]
26     output = torch.cat([O_diag_head, O_merged_tail]) # [L, d]
27     return output
```

---

Table 4: Performance of GPT-4 and 13 community models on the Needle-in-a-Haystack task at various document depths. The document is split into three equal segments: 0-33% depth, 33-66% depth, and 66-100% depth. **Peak Failure Depth** indicates the document depth at which the most test cases failed for each model. Results are reported at the training length for each model.

| Model | $L_{train}$ | HF_PATH | Peak Failure Depth | Acc |
|---|---|---|---|---|
| GPT-4-128K | – | – | 0-33.3% | 100.0 |
| **Trained on open-source data** | | | | |
| TinyLlama-1.3b-1T(ours) | 2k | – | 0-33.3% | 56.6 |
| TinyLlama-1.1b-1T | 2k | TinyLlama/TinyLlama-1.1B-intermediate-step-480k-1T | 0-33.3% | 38.0 |
| TinyLlama-1.1b-3T | 2k | TinyLlama/TinyLlama-1.1B-intermediate-step-1431k-3T | 0-33.3% | 69.8 |
| Pythia-1.4b | 2k | EleutherAI/pythia-1.4b | 0-33.3% | 22.5 |
| OpenLlama-3B | 2k | openlm-research/open_llama_3b | 0-33.3% | 85.0 |
| Llama2-7B | 4k | meta-llama/Llama-2-7b | 0-33.3% | 98.0 |
| Llama3-8B | 8k | meta-llama/Llama-3-7b | 0-33.3% | 99.8 |
| Together-base | 32k | togethercomputer/Llama-2-7B-32K | 0-33.3% | 63.0 |
| LWM-base | 32k | LargeWorldModel/LWM-Text-32K | 0-33.3% | 31.8 |
| Mistral-base | 32k | alpindale/Mistral-7B-v0.2-hf | 0-33.3% | 52.8 |
| Llama3.1-8B | 128k | meta-llama/Meta-Llama-3.1-8B | 0-33.3% | 66.0 |
| Yarn-base | 128k | NousResearch/Yarn-Llama-2-7b-128k | 0-33.3% | 32.4 |
| Yi-6b-200k | 200k | 01-ai/Yi-6B-200K | 0-33.3% | 20.8 |
| Gradient-Llama3-8B | 262k | gradientai/Llama-3-70B-Instruct-Gradient-256k | 0-33.3% | 46.0 |

Table 5: The input format of the Needle-in-a-Haystack (4-Needle) test where the needles are 6-digit numbers and the haystack is Paul Graham Essays (gkamradt, 2023). The needles we use in this work are numbers to exclude the influence by inner-knowledge following previous work (Zhang et al., 2024c; Mohtashami & Jaggi, 2023; Hsieh et al., 2024; Zhang et al., 2024d)

| Haystack Needles Query | There is an important info hidden inside a lot of irrelevant text. Find it and memorize them. I will quiz you about the important information there.\n\n July 2006I've discovered a handy test for figuring out what you're addicted to. Imagine you were going to spend the weekend at a friend's house on a little island off the coast of Maine. There are no shops on the island and you won't be able to leave while you're there. Also, you've never been to this house before, so you can't assume it will have more than any house might.What, besides clothes and toiletries, do you make a point of packing? That's what you're addicted to... One of the magic numbers is 144231. they're going to issue a public report tracking how this specific tranche of money is spent, NFTs are a new territory, and this way of using them is especially new, but I'm excited about its potential. And I'm excited to see what happens with this particular auction, because unlike an NFT representing something that has already happened, this NFT gets better as the price gets higher.The reserve price was about $2.5 million, because that's what it takes for the name to be accurate: that's what it costs to... One of the magic numbers is 543171. you can't expect an idea to be novel to everyone. Any insight that you have will probably have already been had by at least one of the world's 7 billion people. But it's sufficient if an idea is novel to a lot of readers.Ditto for correctness, importance, and strength. In effect the four components One of the magic numbers is 264468. And we were always far ahead of them in features.Sometimes, in desperation, competitors would try to introduce features that we didn't have. But with Lisp our development cycle was so fast that we could sometimes duplicate a new feature within a day or two of a competitor announcing it in a press release. By the time journalists covering the press release got round to One of the magic numbers is 423103. nThere is a founder community just as there's a VC community. They all know one another, and techniques spread rapidly between them. If one tries a new programming language or a new hosting provider and gets good results, 6 months later half of...\n\n What are the magic numbers mentioned in the provided text?\n The numbers are |
|---|---|
| *Answer* | [144231, 543171, 264468, 423103] |

