# OpenReview forum: "Why Does the Effective Context Length of LLMs Fall Short?"
_ICLR.cc/2025/Conference — ICLR 2025 Poster_

### Official Review · Reviewer_QJGa · 2024-11-02

**Soundness:** 2
**Presentation:** 3
**Contribution:** 2
**Rating:** 6
**Confidence:** 3

**Summary:**

This paper discusses how current LLMs use only a portion of their maximum context length due to issues in handling distant information. The authors identify the root cause as a left-skewed distribution in the model's relative positions during pretraining and post-training.

To address this, they introduce a new method called Shifted Rotary Position Embedding (STRING), which adjusts well-trained positions during inference to improve performance without extra training.
Experimental results show that STRING enhances the context length usage of open-source models like Llama 3.1 70B and Qwen2 72B by over 10 points on benchmarks like RULER and InfiniteBench. STRING's effectiveness even surpasses commercial models like GPT-4-128K, Claude 2, and Kimi-chat.

**Strengths:**

The paper is well-structured, with clear organization, understandable tables, and figures, effectively addressing an important topic in LLM development.

The empirical analysis of position frequency on the training data is thorough (However, I recommend providing additional explanation regarding the meaning of the plot in Figure 1 in the Appendix for further clarity) .

The method has been evaluated on/against quite a few LLMs.

**Weaknesses:**

1. The code is missing; the provided GitHub link leads to a 404 error.

2. The analysis in Section 3 is intriguing, but the results are limited to small-sized LLMs. It remains uncertain whether the same findings apply to medium- and large-sized LLMs.

**Questions:**

1. Where can we find your implementation of STRING?

2. Are error bars available for the tables in the experiments? If space is limited, could they be included in the Appendix to give readers a better understanding of the variability in the results?

3. Regarding the improved results shown in the experiments, could you provide more insights into why and how STRING performs effectively? Additionally, are there any scenarios where STRING might not perform as well?

---

> ### Author Response · Authors · 2024-11-22
> **Author Response**
>
> Thank you for a detailed review and valuable suggestions. In response to your questions, our answers are as follows.
>
> > **Q1:** Where to find our implementation of STRING.
>
> **A1:** The link in our manuscript is a placeholder, and it will be replaced in our camera-ready version. The pytorch code of STRING can be found in the submitted supplementary materials. We have also created an anonymous [GitHub page](https://anonymous.4open.science/r/ICLR-example-B26B/README.md) detailing our implementation to ensure transparency and reproducibility.
>
> > **Q2:** Show the variability in the results.
>
> **A2:**  All the experiments run in our paper use greedy search following previous work, and there is no training procedure involved. To show the error bar, we use a top-p sampling strategy with top-p set to 0.7.
>
> | Task | Llama3.1 8B (greedy) | Llama3.1 8B (sampling-1) | Llama3.1 8B (sampling-2) | Llama3.1 8B (sampling-3) | mean | Std Dev |
> |:--|:--|:--|:--|:--|:--|:--|
> | RULER-NIAH | 94.0 | 94.2 | 93.6 | 94.2 | 94.0 | 0.24 |
> | RULER-VT | 88.1 | 88.1 | 88.1 | 88.1 | 88.1 | 0.0 |
> | RULER-Aggregation | 37.6 | 37.8 | 37.2 | 37.2 | 37.4 | 0.25 |
> | RULER-QA | 62.7 | 62.2 | 62.5 | 62.0 | 62.3 | 0.26 |
> | En.Sum | 28.22 | 26.98 | 28.87 | 28.61 | 28.17 | 0.72 |
> | En.QA | 10.20 | 10.06 | 10.34 | 10.24 | 10.20 | 0.10 |
> | En.MC | 70.30 | 69.89 | 71.61 | 69.02 | 70.20 | 0.93 |
> | En.Dia | 19.50 | 20.00 | 20.50 | 20.00 | 20.00 | 0.35 |
> | Retr.PassKey | 100.0 | 100.0 | 100.0 | 100.0 | 100.0 | 0.0 |
> | Retr.Number | 99.89 | 99.83 | 99.92 | 99.84 | 99.87 | 0.03 |
> | Retr.KV | 83.00 | 81.40 | 81.60 | 82.80 | 82.2 | 0.70 |
> | Code.debug | 26.90 | 30.22 | 27.04 | 26.69 | 27.71 | 1.45 |
> | Math.find | 34.87 | 34.73 | 34.84 | 34.82 | 34.81 | 0.05 |
>
> We run the model with top-p decoding 3 times denoted as `sampling-1, sampling-2, sampling-3` and the results are shown in this table.
>
> > **Q3**:  Whether the same findings apply to medium- and large-sized LLMs
>
> **A3:** We agree that conducting experiments with larger models at scales of 7B or 70B parameters could enhance the robustness of our results. However, a 1.3B parameter model is the largest size that our current resources can support for pretraining from scratch.  In fact, most models pretrained with open-source corpus from the community are usually at the scale of GPT-2 or TinyLlama.  Section 3 is also named as `a probing experiment` to avoid any overclaims. The analysis of position frequency in large language models in should be conducted during the pretraining phase and we are unable to use models from big companies. Despite utilizing many acceleration strategies following the popular project TinyLlama, we still require one month to train each model with 16 NVIDIA A100 GPUs .
>
> For large-sized models, we demonstrate the importance of position frequency by results in **Section 4**: by abandoning the use of infrequent positions for gathering distant information, large-sized models like Llama3.1 70B and Qwen2 72B can be greatly improved. This also indicates that the long-context ability of large-sized models is also hindered by infrequent positions.
>
> > **Q4:** More insights about STRING and potential limitations of STRING.
>
> **A4:** The core idea of STRING is to use well-trained positions to gather distant information while maintaining the closest distance to neighboring tokens. In the original ROPE, distant information is gathered by the most infrequent positions, causing the model to overemphasize local dependency rather than long-range dependency. For example, in long-context QA, each token needs to focus on its neighboring tokens to ensure the fluency, so tokens with a distance of 1 are always more important than tokens with a distance of 1K. However, tokens at a distance of 1K are not necessarily more important than those at a distance of 100K. It is possible that the tokens at a distance of 100K are precisely the ones that can be used to infer the answer. Compared to ROPE, STRING is more conducive to gathering such information.
>
> A potential bias of STRING is introducing *non-consecutive position indices*. Taking the position matrix of Llama3.1 as an example,
> |   |   |   |   |   |   |
> |---|---|---|---|---|---|
> | 0 |   |   |   |   |   |
> |  ... | 0 |   |   |   |   |
> | 42K-1 | ... | 0 |   |   |   |
> | W | 42K-1 | ...  | 0 |   |   |
> | ...   | W | 42K-1 |  ... | 0 |   |
> | 86K+W-1 | ...  | W | 42K-1 | ...  | 0 |
>
> the `(42K-1)-th` diagonal is filled with `42K-1` but the `42K-th` diagonal is filled with `128`. However, we have conducted extensive experiments to verify STRING on `22 synthetic and real-world sub-tasks` from RULER and InfiniteBench, and we have not observed any tasks where STRING fails to outperform ROPE. This may suggest that the benefits brought by STRING for current LLMs greatly outweigh the potential bias it may cause.

---

> ### Author Response · Authors · 2024-11-27
> **Follow-up on Review Feedback**
>
> Dear Reviewer QJGa,
>
> Thank you sincerely for taking the time and effort to review our submission.
>
> Below, we address the points you raised:
>
> - The GitHub link currently included in our manuscript serves as a `temporary placeholder` and `will be updated` in the final camera-ready version of our paper.
> - We have added `error bars` to the tables in our experimental section to better illustrate the variability of our results.
> - We have expanded the discussion on STRING, providing `deeper insights` and outlining its `potential limitations`.
> - Our current computational capacity is limited to `16 NVIDIA A100` GPUs and it took us about 2 months to pretrain these models from scratch. Larger models developed by big companies, such as Llama2 7B, require significantly more resources (e.g., Llama2 7B demands about 3.3 million GPU hours). However, the effect caused by infrequent positions on large-sized models can be demonstrated by the improvements from STRING.
>
> As the public discussion phase comes to an end, please let us know if there are any additional questions or concerns we can address to assist with your review. We are more than willing to provide any further information or clarification you might need.
>
> Thank you once again for your valuable feedback.

---

### Official Review · Reviewer_rtHL · 2024-11-02

**Soundness:** 3
**Presentation:** 3
**Contribution:** 2
**Rating:** 6
**Confidence:** 3

**Summary:**

This paper suggests that large language models (LLMs) struggle to handle dependencies over long distances due to the skewed occurrence of relative distance towards shorter ranges while training. The authors propose STRING, a simple method that re-uses more frequently occuring positions for distant tokens, avoiding the use of under-utilized long-distance positions. Through extensive evaluations on long-context benchmarks, they demonstrate that STRING can improve the performance of SOTA models within their pre-trained context windows.

**Strengths:**

1. The paper is clearly written and easy to follow.
2. The proposed method is simple and effective. In particular, it improves the performance of existing LLMs within their pre-trained context window.
3. The proposed method can be easily deployed. STRING does not require any additional training and can therefore be easily applied to existing LLMs. The authors also provide a way to make the method compatible with FlashAttention, enabling fast and efficient inference.

**Weaknesses:**

1. Observation (1) in Section 3 (line 216) is overstated, as training with longer context involves larger computation demands due to the quadratic nature of self-attention. For fair comparison, training efficiency should be compared with the same amount of computation, instead of the number of consumed tokens.
2. STRING looks more like an empirical workaround to avoid using less-frequent positions, rather than a fundamental solution for the problem.

**Questions:**

1. Can this method also be useful for context extrapolation? I understand that the primary focus of the paper is to improve performance within the pre-trained context window, but I believe this is also a large understated benefit of your method.

---

> ### Author Response · Authors · 2024-11-22
> **Author Response**
>
> We are grateful for your valuable review and thoughtful suggestions, and we try to answer your questions as follows.
>
> > **Q1**: Observation (1) in Section 3 (line 216) is overstated.
>
> **A1:** We double-checked our paper and found no claims about training efficiency. The original claim in `line 216` is 'Larger training context window consumes fewer tokens to achieve the same effective context length' and we did not make claims like 'Larger training context window achieves the same effective context length with better training efficiency'. The number of consumed tokens is closely related to the position frequency studied in this work. We have added the following sentence in our updated manuscript (`line 221`): *'But it does not mean larger context training window can save more computation'* for better clarity.
>
> > **Q2**: STRING is not a foundational solution.
>
> **A2:**  We appreciate the concern that STRING may not be classified as a foundational or final solution. However, the primary objective of our work is to highlight a critical issue that contributes to the limited long-context capabilities of current large language models (LLMs).
>
> We introduce STRING with two main goals:
> * STRING is designed as a plugin that can be seamlessly integrated into existing models, delivering significant improvements in their long-context handling abilities.
> * By showcasing STRING's effectiveness, we aim to illustrate the substantial potential in addressing the identified problem.
>
> We hope that the community can make significant strides in enhancing the long-context processing capabilities of LLMs inspired by our analysis and the design of STRING.
>
> > **Q3:** Extrapolation experiments
>
> **A3:** Thank you for your suggestions and we agree that the extrapolation results provide a valuable dimension to our method. Positions exceed the training length can be mapped to the trained interval with STRING.
>
> We verify the extrapolation version of STRING with language modeling and NIAH (4-needle). The test length is set to 1.0x training length to 2.5x training length. STRING achieves a better performance than popular extrapolation methods: NTK, YaRN and ReRoPE. The detailed results are shown in the following tables.
>
> **PG19 Language modeling**
>
> | Model             | 1.0x | 1.5x | 2.0x | 2.5x |
> |-------------------|------|------|------|------|
> | TinyLlama 2k  | 7.91 | --   | --   | --   |
> | + NTK             | 7.96 | 8.00 | 8.63 | 9.62 |
> | + Yarn            | 8.01 | 7.90 | 7.99 | 8.29 |
> | + ReRoPE          | 8.12 | 8.10 | 8.05 | 8.09 |
> | + STRING          | 7.82 | 7.93 | 7.80 | 7.81 |
> | Llama2 4K         | 7.46 | --   | --   | --   |
> | + NTK             | 7.58 | 7.62 | 11.98| --   |
> | + Yarn            | 7.50 | 7.42 | 7.67 | 7.73 |
> | + ReRoPE          | 7.49 | 7.48 | 7.54 | 7.69 |
> | + STRING          | 7.39 | 7.30 | 7.24 | 7.33 |
> | Llama3 8K         | 8.60 | 13.55| --   | --   |
> | + NTK             | 8.67 | 8.92 | 9.72 | 10.32|
> | + Yarn            | 8.63 | 8.44 | 8.16 | 8.05 |
> | + ReRoPE          | 8.68 | 8.60 | 8.28 | 8.03 |
> | + STRING          | 8.30 | 8.21 | 8.04 | 7.85 |
>
> **Needle-in-a-haystack**
> | Model             | 1.0x | 1.5x | 2.0x | 2.5x |
> |-------------------|------|------|------|------|
> | TinyLlama 2k  | 69.8 | 0    | 0    | 0    |
> | + NTK             | 79.8 | 62.0 | 39.0 | 0    |
> | + Yarn            | 88.0 | 70.2 | 49.4 | 35.8 |
> | + ReRoPE          | 77.2 | 50.2 | 34.8 | 22.8 |
> | + STRING          | 97.2 | 88.4 | 48.6 | 40.0 |
> | Llama2 4K         | 98.0 | 0    | 0    | 0    |
> | + NTK             | 98.6 | 50.0 | 20.2 | 0    |
> | + Yarn            | 98.0 | 61.8 | 55.0 | 15.2 |
> | + ReRoPE          | 98.6 | 55.4 | 43.0 | 5.0  |
> | + STRING          | 100.0| 84.2 | 58.4 | 27.6 |
> | Llama3 8K         | 99.8 | 0    | 0    | 0    |
> | + NTK             | 100  | 98.2 | 9.0  | 0    |
> | + Yarn            | 100  | 93.0 | 66.8 | 48.2 |
> | + ReRoPE          | 99.6 | 95.6 | 75.4 | 59.4 |
> | + STRING          | 99.6 | 98.2 | 79.0 | 74.4 |

---

> ### Comment · Reviewer_rtHL · 2024-11-26
>
> Thank you for the clarifications and for your efforts on the extrapolation experiments. My concerns are resolved, and I believe STRING will provide valuable insights for future improvements on position embeddings.

---

> > ### Author Response · Authors · 2024-11-26
> > **Thank you for your response**
> >
> > Thank you for taking the time to read our response. We are delighted to hear that your concerns have been addressed and that you recognize STRING's potential for future improvements in position embeddings. Your insightful comments have significantly improved our paper and made it clearer.

---

### Official Review · Reviewer_v29A · 2024-11-03

**Soundness:** 4
**Presentation:** 4
**Contribution:** 4
**Rating:** 8
**Confidence:** 4

**Summary:**

This paper reveals the possible reasons why current LLM can't maintain good performance with long-context task even the training has involved such corpus. According to the analysis on the left-skewed frequency-distribution, the paper proposes a training-free method (STRING) to mitigate the position bias. By replacing infrequent long-range position index to frequently trained positions during inference, their method enhances the long-context capability without requiring additional training. The experiment results on multiple long-context benchmarks demonstrate the effectiveness of STRING.

**Strengths:**

1. The exploratory analysis about the left-skewed frequency-distribution of position index is insightful, and can help the community understand the potential reasons why the long-context problem exists in current methods.

2. The proposed method is intuitive and simple to implement, which seems effective simultaneously.

3. The presentation of the paper is very clear,  and the analysis together with the method description are easy to follow,

**Weaknesses:**

1. The parameter S and W needs to be set manually, which maybe improved by an automatic algorithm to obtain a better performance.

**Questions:**

It is shown that the most distant position S indexes can be ignored during inference to obtain a better performance, have you conducted investigation whether it is the same case for training? Can decreasing the longest context window in training directly yields a better performance in inference?

---

> ### Author Response · Authors · 2024-11-22
> **Author Response**
>
> Thank you for a detailed review and valuable suggestions!
>
> > **Q1: Automatically setting S and W.**
>
> **A1:** Thank you for the suggestion! In our ablation study (see Figure 7), we find that the performance variation remains minimal when W ranges from 32 to 512. Automatically setting W may not significantly impact the results.
> However, for S, its value is influenced by the number of well-trained positions in the model. In our experiments, we set S to $\frac{1}{3}$ of the training length for all tasks and models to facilitate reproducibility and follow-up by the community.
> Automatically setting S could potentially yield better results, such as through a grid search on synthetic data. We plan to explore this in future work.
>
> > **Q2: Ignoring distant positions during training.**
>
> **A2:** In our training setting, we use Llama2 4K and the latest Llama3.1 128K as the base model. For Llama2 4K, we use conversations from ShareGPT and set the training length to 4K. For Llama3.1 128K, we train with a popular long-context SFT dataset, ChatQA-2[1] and set the training length to 128K. We train the models for one epoch.
>
> The table below shows the results on RULER, where -FT indicates the finetuned version of STRING.
>
> | Model                | NIAH | VT   | Aggregation | QA   | Avg (13 tasks) |
> |----------------------|------|------|-------------|------|----------------|
> | Llama2-ROPE          | 96.9 | 89.7 | 84.8        | 49.7 | 87.2           |
> | Llama2-STRING        | 97.0 | 90.5 | 89.6        | 52.5 | 88.5           |
> | Llama2-STRING-FT     | 97.5 | 91.8 | 86.1        | 60.6 | 89.6           |
> | Llama3.1-ROPE        | 92.6 | 70.4 | 36.2        | 58.8 | 77.0           |
> | Llama3.1-STRING      | 94.0 | 88.1 | 37.6        | 62.7 | 80.0           |
> | Llama3.1-STRING-FT   | 94.5 | 86.6 | 35.9        | 64.5 | 80.2           |
>
> The table below shows the results on RULER, where `STRING-FT` indicates the finetuned version of STRING. For Llama2, our test length is set to 4K, and for Llama3.1, it was 128K. The results show that continual training can lead to more performance gain. On Llama2 4K, the improvements are significant, but on Llama3.1, the benefits are less pronounced, possibly due to data quality issues.
>
> In addition to simplicity and flexibility, our primary reason for not including the training settings in the initial version is to exclude the impact from data. The training-free improvements are clearly attributed to the exclusion of infrequent positions rather than the use of additional training data.
>
> [1] ChatQA 2: Bridging the Gap to Proprietary LLMs in Long Context and RAG Capabilities

---

> > ### Comment · Reviewer_v29A · 2024-11-25
> >
> > Thanks for the effort. The response resolves my concern.

---

> > > ### Author Response · Authors · 2024-11-26
> > > **Thank you for your response**
> > >
> > > Thank you for taking the time to read our response. We would like to express our gratitude once more for your detailed review and constructive feedback.

---

### Official Review · Reviewer_SwNh · 2024-11-10

**Soundness:** 3
**Presentation:** 3
**Contribution:** 3
**Rating:** 6
**Confidence:** 4

**Summary:**

The authors present STRING (Shifted Rotary position embedding) which proposes to shift the position relevance of tokens during inference. With STRING, the authors are able to demonstrate improved performance of 10+ points on RULER, and IB for LLAMA 70B. The authors further train a tiny-llama 1.3B model to demonstrate the effectiveness and compare STRING with SOTA models. The authors also discuss the implementation of STRING with modification to flash attention.

**Strengths:**

Thank you for submitting your paper to ICLR. I thoroughly enjoyed reading your paper, especially it is very well written.
1. The paper observes "left-skewed position frequency distribution" - a bias in the position indices required to answer a relevant question being towards the shorter end. For example, on SlimPajama, the authors observe that the frequency of position indices used to model relationships between distant tokens (distances
≥1024) is less than 20%, and for even longer distances (≥1536), it drops below 5%.
2. The authors also demonstrate interesting findings, such as "larger training context window consumes fewer tokens to achieve the same context length". Although it's only been tested on tinyllama (1.3B) so weather scaling laws hold is an open problem.

**Weaknesses:**

1. The authors acknowledge in part (3) of 4.1 that "Applying Eq. 3 disrupts the model’s ability to capture
local relationships because it alters the relative positions between neighboring tokens" but I'm not fully convinced if the heuristic of "W" is sufficient to allevate this concern. While emperically the experiments suggest so, I'd have liked to see a slightly more detailed study on the impact of altering the relative positions between neighboring tokens.

**Questions:**

The results in the paper for the 1.3B Tiny-llama and the commercial models are impressive on the long-context benchmarks. However, assuming the "system" to mean the LLM and STRING combination will be used together, I'd also like to see benchmarks for short-context lengths - especially around coding (MBPP, Humaneval), tool-use (BFCL), and Reasoning (GSM-8K, MATH). If not for all, atleast for 1/2 models to reason if there are any trade-offs. I'm especially curious for coding and tooling benchmarks, since they need to adhere to syntax, which might be compromised by the re-positioning.

Since you have implemented STRING, what is the latency overheads (and impact on throughput) vis-a-vis vanilla RoPE?

---

> ### Author Response · Authors · 2024-11-22
> **Author Response (Part 1)**
>
> We appreciate your thorough review as well as insightful comments, and try to answer your questions as follows.
>
> > **Q1**: Detailed explanation of how we protect local relationships.
>
> **A1**:  To illustrate this, consider the relative matrix of Llama3.1 128K,
>
> |   |   |   |   |   |   |
> |---|---|---|---|---|---|
> | 0 |   |   |   |   |   |
> |  ... | 0 |   |   |   |   |
> | 42K-1 | ... | 0 |   |   |   |
> | W | 42K-1 | ...  | 0 |   |   |
> | ...   | W | 42K-1 |  ... | 0 |   |
> | 86K+W-1 | ...  | W | 42K-1 | ...  | 0 |
>
>  where the value on the 42K-th diagonal is set to W instead of 42K (using ROPE) to achieve a shifting effect. This means each token treats the token 42K positions away as if it's at a distance of W.
> In an extreme case, if W=0, the current token would treat the token 42K positions away as itself. As we know, LLMs rely heavily on neighboring tokens to maintain fluency. If W is too small, the model can not correctly focus on these nearby tokens. By setting W=128, we ensure that relative positions 0-127 are unique and assigned to neighboring tokens. This maintains fluency while enhancing long-context performance.
>
>
> > **Q2**: The impact of altering the relative positions between neighboring tokens.
>
> **A2**:  As explained in A1, when W is set to a small value, it disrupts the relative positions of neighboring tokens. We investigate this effect by testing three versions of Llama with different training lengths, and we vary W from 0 to 128. The table below shows the results on the NIAH task:
> | Model         | W=0 | W=2 | W=4  | W=8  | W=16 | W=32 | W=64 |
> |---------------|-----|-----|------|------|------|------|------|
> | Llama2 4K     | 0   | 0   | 20.6 | 95.2 | 92.4 | 100  | 100  |
> | Llama3 8K     | 0   | 0   | 10.2 | 90.0 | 88.0 | 100  | 97.6 |
> | Llama3.1 128K | 0   | 0   | 16.0 | 95.2 | 90.6 | 89.8 | 96.4 |
>
> In this table, we observe that the **nearest 8 tokens** are the most crucial for fluent generation. When W > 8, there is no significant degradation. Setting W to a larger value, such as 128, usually results in more stable outcomes.

---

> ### Author Response · Authors · 2024-11-22
> **Author Response (Part 2)**
>
> > **Q3**: Potential side-effects caused by the re-positioning on short-context benchmarks
>
> **A3 Part-I:  STRING will produce the same results as RoPE on short-context benchmark.**
>
> Considering the Pseudocode Code for Llama3.1 128K in our implementation:
> ```python
> l = input_length
>
> L = 128K # training length
> S = 42K # L // 3, shifted size
>
> diag_attn = flash_attn(q, k, v, sliding_window=S)
> if l <= S:    # for short context, l<= S is always True
>     return diag_attn
> else:
>     do shifted self-attention
>     merge sliding window attention and self-attention
>     return merged_attn
> ```
> When the input length $l < S$, the sliding window attention calculates attention within the neighboring $S$ tokens. In the short-context setting, we always have $S \gg l$, and the sliding window attention calculates full attention for the short sequence without shifting position indices. Results on the long-context coding dataset `code.debug` which requires the model to adhere to syntax are shown in **Table 3** of our manuscript. STRING is able to yield more than a `3-point` score improvement on Llama3.1 8B/70B.
>
> **A3 Part-II: additional discussion.**
>
> Since short-context input does not involve infrequent positions, there is also no motivation to re-order positions in this scenario. However, we agree that studying the side effects of re-positioning on short-text tasks is a very interesting idea.
>
> We update the code so that the shifted size S is now determined by the input length rather than the training length.
>
> ```
> l = input_length # can be a small value
>
> S = l//3  #  shifted size determined by input length
> N = 2*l//3
>
> diag_attn = flash_attn(q, k, v, sliding_window=S) # l // 3
> q = apply_shifted_pos(q) # apply shifted position ids for q
> shifted_attn = flash_attn(q[-N], K[:N], V[:N]) # shifted-self-attention (the bottom-left triangle)
>
> output = merge(diag_attn, shifted_attn)
> ```
> If we still keep W = 128, it will result in the following position matrix:
> |           |   |   |   |   |   |   |
> |-----------|---|---|---|---|---|---|
> | 0         |   |   |   |   |   |   |
> |    ...       | 0 |   |   |   |   |   |
> | $\frac{l}{3}$-1     | ...  | 0 |   |   |   |   |
> | 128       |  $\frac{l}{3}$-1  | ...  | 0 |   |   |
> |    ...       | 128 |  $\frac{l}{3}$-1  |  ... | 0 |   |
> | $\frac{2l}{3}$+127  | ...  | 128 |  $\frac{l}{3}$-1  | ...  | 0 |
>
> When the input length $l=384$, we have $\frac{l}{3}=128$, which has the same effect as RoPE. Whether $\frac{l}{3}<128$ or $\frac{l}{3}>128$, re-positioning occurs. We study the side-effect of the re-positioning with Llama2 and Llama3.1 on `HumanEval` and `GSM8K`.
>
> **GSM8K**
> |                  | 0-shot ($\frac{l}{3}=21$) | 1-shot ($\frac{l}{3}=62$)  | 4-shot ($\frac{l}{3}=175$) | 16-shot($\frac{l}{3}=976$) |
> |------------------|-----------------|-----------------|------------------|-------------------|
> | Llama2           | 12.35           | 13.08           | 17.51            | 19.25             |
> | + re-positioning | 4.76            | 4.89            | 17.43            | **19.34**             |
> | Llama3.1         | 55.72           | 65.73           | 69.45            | 68.61             |
> | + re-positioning | 34.56           | 54.05           | **69.57**           | 67.24             |
>
> **HumanEval**
> |                  | 0-shot($\frac{l}{3}=49$) | 1-shot ($\frac{l}{3}=113$) | 4-shot ($\frac{l}{3}=277$)| 16-shot($\frac{l}{3}=872$) |
> |------------------|-----------------|------------------|------------------|-------------------|
> | Llama2           | 12.19           | 13.97            | 13.41            | 17.68             |
> | + re-positioning | 0.0             | 12.8             | 12.19            | 15.24             |
> | Llama3.1         | 54.26           | 59.14            | 56.70            | 54.39             |
> | + re-positioning | 44.82           | 54.04            | 48.78            | 53.51             |
>
> Overall when  $\frac{l}{3}<128$, the performance decline is most noticeable and as the sequence length increases, the negative effect of re-positioning becomes smaller. Only in the GSM8K with 4-shot and 16-shot settings, re-positioning shows a slight advantage. Compared with math problems, in coding tasks, using re-positioning often leads to worse outcomes as it may disrupt code syntax as you mentioned.
>
> > **Q4**: Latency overheads.
>
> **A4**: We show the inference speed and GPU memory usage of STRING and RoPE in Appendix Figure 9. The throughput and GPU memory consumption are negligible compared to RoPE. The current implementation of STRING is based on calling Flash-Attention twice, and there is room for optimization from the kernel.
>
>
> **Truncated comment**: Thank you again for your kind words and interesting insights. We are pleased to hear that you enjoyed reading our paper. It seems that `Weakness 1` was truncated in OpenReview. Could you please repeat it in a later comment?

---

> ### Author Response · Authors · 2024-11-27
> **Follow-up on Review Feedback**
>
> Dear Reviewer SwNh,
>
> We sincerely appreciate the time and effort you have devoted to providing thoughtful and constructive feedback on our work.
>
> Here is a **summary** of our response:
>
> - A detailed explanation of how the local value `W=128` works and why positions `0~127` are only assigned to neighboring tokens.
> - When distant tokens are assigned relative positions `less than 4`, the model will collapse.
> - STRING produces `the same results` as RoPE on short-context input because only the sliding window attention function is invoked.
> - Side effects caused by` re-positioning` on math and coding tasks.
>
> As the public discussion phase is nearing its conclusion, we wanted to kindly follow up to see if there are any additional questions, concerns, or points that we could clarify or address to further assist with your review process. We are more than happy to provide any additional information or details you might need.
>
> Thank you once again for your valuable comments.

---

> > ### Comment · Reviewer_SwNh · 2024-11-28
> >
> > Thank you for your detailed explanation. My concerns have been addressed. What I find most interesting is that the nearest 8 tokens are the most crucial for fluent generation across 4, 8, and 128K tokens.

---

> > > ### Author Response · Authors · 2024-11-28
> > > **Thank you for your response**
> > >
> > > Thank you very much for your thoughtful feedback and for acknowledging that your concerns have been addressed. We are pleased to hear that you found the finding interesting.  We sincerely appreciate your thoughtful engagement with this work and the valuable insights you have provided.

---

### Author Response · Authors · 2024-11-22
**General Response**

Dear reviewers,

We are deeply grateful for your insightful feedback and valuable suggestions. Your comprehensive reviews have provided us with essential guidance to enhance our work

We also wish to express our appreciation for your recognition of the strengths of our work, including:

* Insightful analysis on the left-skewed frequency distribution of position indices which potentially hinders the long-context ability of current LLMs (`SwNh`, `v29A`, `rtHL`, `QJGa`)
* The proposed positional encoding STRING is effective with extensive  evaluation (`v29A`, `rtHL`, `QJGa`)
* The paper has a clear organization. The analysis together with the method STRING are easy to follow (`v29A`,` rtHL`, `QJGa`)

Below is a summary of the key updates made to the manuscript:

- Enhanced explanations (Appendix A.1):
    - How we utilize $W$ to preserve the relationships between neighboring tokens.
    - Why STRING produces results consistent with RoPE in short-context settings.
- Additional results:
    - Results of STRING in a training setting (Table 2).
    - Results of STRING in a context extrapolation setting (Appendix Table 4,5).
    - Error bars to the tables in our Experiments section (Appendix Table 6)

---

### Meta-Review · Area_Chair_J4FD · 2024-12-18

**Metareview:**

This paper proposes a simple and effective method that re-uses more frequently occuring positions for distant tokens via analyzing skewed distribution of LMs.

All reviewers agree the contributions of this paper such as interesting findings, in-depth analsysi, effective method, and promising results. Despite some minor concerns such as lack of experiments at scale, this paper can contribute to LLM community in terms of context length and position embedding.

So, AC recommends accepting this paper.

**Additional Comments On Reviewer Discussion:**

The initial scores were 6, 8, 6, and 6 with some minor concerns raised by the reviewers.

During the rebuttal period, the authors successfully addressed most issues, and all reviewers kept their score.

---

### Decision · Program_Chairs · 2025-01-22

Accept (Poster)